# Aggregate Electric Vehicles in Demand Side for Ancillary Service

**Xianglu Liu [1,\*], Xianglong Li [1], Haiyang Chen [2], Wenbin Zhou [1], Zhou Sun [1] and Hao Xu [3]**

[1] State Grid Beijing Electric Power Research Institute, Beijing 100075, China; lxl0_0@163.com (X.L.); wenbinchow@126.com (W.Z.); sunzhou0812@163.com (Z.S.)
[2] State Grid Beijing Electric Power Company, Beijing 100051, China; chy0713@163.com
[3] State Grid Beijing Changping Electric Power Supply Company, Beijing 102299, China; beyondsai@126.com
\* Correspondence: liuxianglu28@163.com

**Abstract:** This paper investigates the win-win commercialization mode of aggregating electric vehicles (EVs) in demand side for ancillary service. We have conducted a half-year-long incentive verification experiment covering 10,066 electric vehicle owners in Beijing. Based on the experimental results, we develop an incentive-based mechanism that enables electric vehicles to participate the wholesale capacity market through an aggregator. The aggregator, which is held by charging service operators can make a profit by designing a smart pricing policy. In this process, not only the electric vehicle owners but also the utility can gain benefits.

**Keywords:** EV (electric vehicle); reserve capacity market; aggregator; price policy

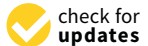



## 1. Introduction

With the development of renewable energy resources, the grid calls for additional flexibility, e.g., in the form of reserves to ensure that electricity generation equals to load. In principle, flexible load such as EVs can take comparable measures that can be taken by conventional generators. Accordingly, it brings conceptual opportunities for these energy resources to provide reserve services [1].

With the advances in battery technologies, there has been a growing interest in finding new and profitable applications of EVs. The existing study has proved that EVs can boost the end use of renewable energy supplies [2], reduce the risk of distribution outages [3] and improve grid reliability by participating in ancillary services [4,5].

In the meanwhile, the penetration rate of electric vehicles is expected to continue increasing rapidly. Compared with large centralized storages owned by the grid operator, EVs which in demand side exhibit several unique advantages:

In the technique level, EVs can response to local variation of the grid quickly through flexible scheduling in a distributed manner. In economics, the large battery investment costs can be dispersed among end users. From incentives, EVs would like to participant in electricity market for additional financial returns.

However, it is challenging for the grid operator to directly control large numbers of EVs, which have their own objectives and constraints. What is more, the reserve capacity market has strict rules on participants, such as the requirements of minimum capacity and operation time, which prevent individual end-users to directly participate. Therefore, an aggregator is necessary to act as an interface between end-users and the grid.

Many related works have focused on an aggregator managing a large number of EVs [6–8]. Ref. [6] proposed a cooperative charging strategy to minimize the social cost. To tackle the potential conflicts between charging needs and the provision of grid services, the algorithm developed in [7] maximizes profits of the aggregator, while providing additional services such as regulation and peak load shaving. Ref. [8] designed a decentralized method to yield optimal scheduling for regulation services.

When it turns to the interaction between aggregator and grid, many related works investigate market participation of the aggregator. In wholesale market, the aggregated EVs are assumed to be price-taker, i.e., not large enough to be price-maker. Refs. [9,10] explored the problem of bidding up/down regulation or spinning reserves for aggregated V2G capable EVs.

In the perspective of the aggregator, it is essential to overcome the predisposition of end-users to remain inactive. Although the incentive issue has been widely considered in literatures concerning demand response (DR) [11–13], few literatures explore the incentive-based mechanism for aggregating EVs.

The optimal market participation is not an easy task for an aggregator. All these above works rely on the assumption that the aggregator exerts full control of EVs as long as they are connected to the power line. That means the aggregator decides whether and when to participant into market for end users. In this way, end users' autonomy in decision-making is lost. On the other hand, when reserve services are required by the grid, in fact, it is the grid, not the aggregator, that has the authority to operate EVs for reserve services.

Since aggregator is benefit-oriented, it will turn to overuse EVs regardless the degradation of battery or individual interests. In addition, lots of works suppose a perfect hypothetical market environment, in which storages can provide reserve services indistinguishably compared with generators [14,15]. In reality, bids placed on wholesale electricity markets must adhere to certain minimum size and operation duration [16]. So, it is unreasonable to ignore these constraints in both volume and time scale.

This paper focuses on exploiting the fast-responding ability of massive EVs for ancillary service of smart grid, in order to address the issue of safe operation of the grid under fast variations of both generation and demand. The key idea is to introduce aggregators who aggregate spare capacity of EVs from end users and sell to utility company, then the utility company can use the capacity in real time. The research involves three types of entities, i.e., end user, aggregator and utility company, two markets, i.e., end user procurement market and ancillary wholesale market. The output of this paper is helpful to address two scientific questions: the stable operation of grids under deep penetration of massive uncertain generations and demands, and the economic value and marketization of massive EVs. The contributions of this paper have three aspects. First, from the engineering practical level, we collected charging information, analyzed over 10,000 user's charging behavior, estimated incentive response and load changes under different prices through a 4-month incentive verification experiment in Beijing. The results verify the incentive effect of price on demand, and provide a dependable practical basis for the user's model in the theoretical analysis. Second, we develop an incentive-based mechanism that enables electric vehicles to participate the wholesale capacity market through an aggregator. That is, the aggregator acts as a price-maker, will buy capacities from end-users, assemble them and then sell to the wholesale market. Third, the profit maximization problem of the aggregator is formulated and solved. While bids in capacity market must meet the minimum bidding size and operating duration, the scheduling and operation of flexible electric vehicle load can be performed in a short-time scale. We design a smart pricing policy, which not only lowering the cost but also fully utilizing fragmented resources in both volume and time scale. Then the proposed model can be transformed into a mixed quadratic linear program (MIQP) by jointly using KKT condition, the disjunctive constraints and the primal-dual approach. Finally, the simulation results demonstrate the proposed model and method is valid.

The remainder of this paper is organized as follows. We describe incentive verification experiment of private EVs in Beijing in Section 2, and set up the profit maximization problem of the aggregator and end users in Section 3. Then the nested optimization problem is transformed into a solvable MIQP by using some mathematical methods. The detail is described in Section 4. Simulation results are exhibited in Section 5, and we conclude in Section 6.

## 2. Incentive Verification Experiment

The incentive verification experiment covers 10,066 electric vehicle owners in 962 housing estates of Beijing. This experiment relying on the YOUYICHONG, which is operated by the subcompany of Beijing electric power company took 14 days as a cycle and lasted for 7 cycles from 11 September to 12 December 2020. Several incentive measures, such as free insurance, first-charging reward and electricity price subsidy are tested. The subsidy for electricity prices varied from 0.1 ¥/kWh to 0.4 ¥/kWh by stages.

When the user selects the orderly charging mode, during the regulation period, the charging pile will start charging at 2.8 KW. During 12:30–16:00 noon and 0:30–7:00 at night, the charging pile will increase the charging power to 7 KW automatically.

As shown in Table 1, among the 10,066 users who participated in the test, 9656 users actually started charging, of which 3867 users participated in orderly charging, accounting for 40.05% of the total charging users. The total number of charging times was 191,600, of which 28,100 were orderly charging, accounting for 14.67% of the total. The accumulated charging power is 4,193,500 kWh, of which the orderly charging power is 623,100 kWh, accounting for 14.86% of the accumulated charging power. The result shows that EV owners respond positively to price subsidy among all the incentive measures.

**Table 1.** Incentive verification experiment results.

| Index | Charging Users | Orderly Charging Users | Charging Times | Orderly Charging Times | Charging Power (kWh) | Orderly Charging Power (kWh) |
|---|---|---|---|---|---|---|
| Total | 9656 | 3867 | 191,600 | 28,100 | 4,193,500 | 623,100 |
| Proportion | / | 40.05% | / | 14.67% | / | 14.86% |

Table 2 shows user participation enthusiasm corresponding to the reward in every stage. The number of users who participate in orderly charging gradually increases as both the average reward and price subsidy go up. When the subsidy increases from 0.2 ¥/ kwh in the 5th stage to 0.3 ¥/ kwh in 6th stage, the proportion of orderly charging users increases from 34.07% to 47.01%. When the subsidy is above 0.15 ¥/kWh, it is attractive to EV owners to keep active in electric grid. It can be expected that if the subsidy increases, the enthusiasm of users will improve. Experimental result shows that users are most sensitive to price, so it is feasible to aggregate EVs by price incentive to provide ancillary service market in the future.

**Table 2.** User participation enthusiasm.

| Stage | Average Reward ¥ | Price Subsidy ¥/kWh | Response Users/Users | Response User Rate % | New User Rate % | Quit User Rate % |
|---|---|---|---|---|---|---|
| 1 | 2.37 | 0.1 | 804/5838 | 13.77% | / | / |
| 2 | 4.9 | 0.1 | 1739/7737 | 22.48% | 77.94% | 4.40% |
| 3 | 5.23 | 0.1 | 1589/7379 | 42.50% | 21.27% | 30.29% |
| 4 | 5.27 | 0.15 | 2075/7932 | 26.16% | 44.05% | 17.52% |
| 5 | 8.82 | 0.2 | 2519/7393 | 34.07% | 35.22% | 15.89% |
| 6 | 13.72 | 0.3 | 2547/5418 | 47.01% | 17.77% | 16.66% |
| 7 | 18.16 | 0.4 | 3338/6823 | 48.92% | 44.86% | 17.98% |

The effects of aggregated load regulation are also remarkable as shown in Figure 1. With the increase of incentive subsidies (from 0.1 ¥/kWh for the 2nd stage to 0.4 ¥/kWh for the 7th stage), the effect of peak shaving and valley filling is remarkable. The daily peak load shifted 2700 kW, and the peak period is delayed for 1 h.

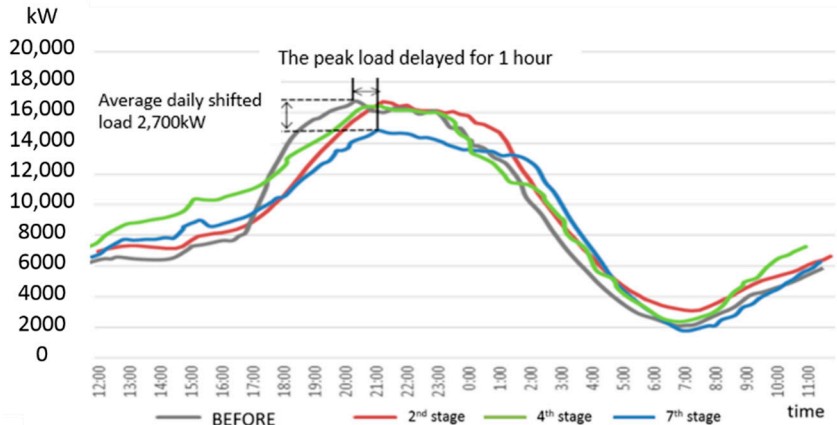

**Figure 1.** The effects of aggregated load regulation.

## 3. Thematic Model Formulation

### 3.1. Market Environment

A subset of these ancillary services can be procured through market-based mechanisms [17], namely, regulation reserve, spinning reserve and non-spinning reserve. Fundamentally speaking, these services with the same effect of supporting system reliability are separated by different response time. Energy storage resources such as EV have been allowed to provide all these ancillary services because of its ability of rapid response and accurate control. Therefore, this paper is not limited to a certain type of products. At the engineering level, EVs are connected with the charging pile when charging. The charging pile is connected to the charging operation platform through the wireless communication module to realize interconnection with the power grid. It can respond to the power grid regulation signal in real time and receive instructions for power control. With the development of orderly charging and V2G, load aggregation and power management have been realized in application.

Because the bids for energy and ancillary services are mutually exclusive, there exits an additional wholesale capacity market in most power systems where the reserve capacities are gathered day-ahead before operating time. The wholesale capacity market has the following characteristics. One is that energy storage equipment e.g., EV is allowed to participate in the market bidding without difference. The other is that industry norms are fully considered. This market has strict market access to ensure the stable operation of the power grid. The service provider can be compensated for ancillary service provision beforehand as long as the capacities allocated to day-ahead market and reserve capacity market do not overlap. As the reserve provision must be reliable, the participants can play a role in reserve capacity market only if some market standards, such as the minimum bidding size or operating interval are obeyed. Thus, in real time, the grid can have the authority to call for these reserve services dealing with the system imbalance or emergency.

Consider end users in a set own private electric vehicle in this system. In addition to the value of transportation, they are willing to resell the stored energy for reserve capacity when the EV remains idle. However, end users are not allowed to directly participant ancillary market, due to their small capacities and fragmented available time.

As a result, the aggregator, which is an intermediary agent between the wholesale market and EV owners, is introduced to coordinate large numbers of these distributed resources to submit regulation capacity in existing capacity markets. In completely competitive market, the market price is determined by the marginal price. Although there are multiple participants in the markets, such as other aggregators. In this paper, we are particularly interested in the behavior of one aggregator. The profit maximization problem of both the aggregator and end users will be set up as follows.

### 3.2. Profit Maximization of Aggregator

In general, it has certain requirements for the capacity scale and continuous operation time. In this paper, we set the capacity scale as 1 MW, and the continuous operation time as 4 h, referring to the requirements of central Europe [1,18]. Consider a future horizon of interest (4 h), which is evenly divided into 16 short time slots (one slot for 15 min). Let $I$ denote the set of end users. Consider an end user $i$ in the smart grid, who is equipped with EV. The user is able to offer the control authority of EV at short slot $h$ to aggregator for reserve services. However, when reserve is authorized at h, the end user loses the control of EV at that slot. In the meanwhile, the power capacities are also specified. The task of the aggregator is to collect reserve capacity of EVs from an individual end user $i \in I$ at purchasing price, and then trade the total capacity at market price in wholesale capacity market. Notice that the purchasing price setting lies on the shorter time scale $h$ owing to EV's feasible adjustment ability, and the bidding size optimization is at the longer time scale $H$.

To minimize the purchasing cost, the aggregator decides its purchasing price $P = \{p_h^r \mid h \in H\}$, while considering the reactions $\{r_{i,h}\} \in S(p_h^r)$ from end users.

The cost of aggregator is:

$$Cost = \sum_{h \in H} \sum_{i \in I} p_h^r r_{i,h} \tag{1}$$

In return, the aggregator decides to optimize reserve bidding size $R$ in ancillary market so as to maximize his income. At $h$ the aggregator owns the authority to operate the aggregated reserve $\sum_{i \in I} r_{i,h}$. However, according to current market rules, the aggregator must submit a constant reserve every $H$ slot to the market, which can be regarded as the minimum capacities in $H$ slots. That is:

$$R = \min\{\sum_{i \in I} r_{i,1}, \sum_{i \in I} r_{i,2}, \ldots, \sum_{i \in I} r_{i,H}\} \tag{2}$$

(2) are complicated by minimize function. To make it more clear, the feasible set of aggregator's bidding size can be summarized as

$$0 \leq R \leq \sum_{i \in I} r_{i,h}, \forall h \tag{3}$$

The aggregator will be paid for $R$ reserve at market price Pr.

Thus, the total income is:

$$\text{Re}v = R\text{Pr} = R\text{Pr}(R) \tag{4}$$

Although there are multiple participants in the market, such as other aggregators, we are particularly interested in the behavior of this aggregator. We assume a residual demand curves of reserve are known to the aggregator and characterized by monotonically decreasing functions. In general, the form of inverse demand function can be assumed to be linear. That is,

$$\text{Pr}(R) := k - nR \tag{5}$$

where $k > 0, n > 0$.

The profit maximization problem for aggregator is:

$$\begin{aligned} &\max_{P,R} \text{Re}v - Cost \\ &s.t.\ 0 \leq R \leq \sum_{i \in I} r_{i,h}, \forall h \\ &\quad \{r_{i,h}\} \in S(p_h^r), \forall h \end{aligned} \tag{6}$$

### 3.3. End User's Problem

The end users are price sensitive, so they will decide their optimal response pattern $r_{i,h}$ to a given purchase price sequence $P$. Based on microeconomic theory, aggregator would receive more reserve by providing a higher purchasing price.

First, the following power limitations should be satisfied:

$$0 \leq r_{i,h} \leq P_i^m, \forall h, \forall i : \lambda_{i,h} \tag{7}$$

$$0 \leq q_{i,h}^d \leq P_i^m, \forall h, \forall i : \rho_{i,h} \tag{8}$$

$$0 \leq q_{i,h}^c \leq P_i^m, \forall h, \forall i : \delta_{i,h} \tag{9}$$

where $P_i^m$ is the maximum discharging and charging power.

The dynamics of SoC is,

$$x_{i,h} = x_{i,0} + \sum_{t=1}^{h} (\alpha q_{i,t}^c - \frac{q_{i,t}^d}{\beta}), \forall h \tag{10}$$

where $x_0$ is the initial energy stored in the storage, and $\alpha, \beta \in (0,1)$ are the charging and discharging efficiency. Then the storage constraint for each end user can be bounded as

$$\frac{r_{i,h}}{\beta} \leq x_{i,h} \leq C, \forall h, \forall i : \gamma_{i,h}, \varphi_{i,h} \tag{11}$$

Clearly, objective function of end user is:

$$U_i = \sum_{h \in H} p_h^r r_{i,h} - u_{i,h} - p_h^e (l_{i,h} + q_{i,h}^c - q_{i,h}^d) \tag{12}$$

where $u_{i,h}$ denote his lost benefit of self-scheduling in period $h$. In general, $u_{i,h}$ is assumed to be nondecreasing and convex. For simplicity, we choose the quadratic function as disutility function.

Therefore, the objective of end user is to maximize the revenue in $H$, by deciding $\left\{ r_i, q_i^d, q_i^c \right\}$ as follows,

$$\max_{r_i, q_i^d, q_i^c} U_i \\ s.t. (7) - (9), (11) \tag{13}$$

As the purchase prices have already been fixed when end users make decisions, the end user problems render quadratic programming. The feasible set can be presented as:

$$S(p_h^r) = \left\{ r_{i,h}, q_{i,h}^d, q_{i,h}^c \right\}, \forall i, \forall h = \arg \max_{r_{i,h}, q_{i,h}^d, q_{i,h}^c \in F} U_i, \forall i \tag{14}$$

where $F$ can be defined as:

$$F = \left\{ r_{i,h}, q_{i,h}^d, q_{i,h}^c \middle| (7) - (9), (11) \right\} \tag{15}$$

There are some difficulties in obtaining a solution to this problem. The profit maximization problem has a bilevel structure where the aggregator's decisions of purchasing price and bidding size strategy depend on the upper level optimization, user's optimal response at the lower level.

Fundamentally speaking, user's problems are nested in the aggregator's problem, which making it challenging to solve directly. Thus, the remainder of the paper will focus on obtaining a solution to this proposed model.

## 4. Solution

Karush–Kuhn–Tucker (KKT) conditions-based reformulation is a powerful method to transform the nested optimization of the lower-level into explicit constraints. If a convex optimization problem with differentiable objective and constraint functions satisfies Slater's condition, then the KKT conditions provide necessary and sufficient conditions for optimality [19]. Therefore, the bilevel problem can be reformulated as a single-level optimization problem. As one can notice, the user's optimization scheme outlined in Section 3.3 is convex with all linear constraints, which satisfies the KKT reformulation. The KKT optimality condition of end user's problem can be represented:

$$0 \le (P_i^m - r_{i,h}) \perp \lambda_{i,h} \ge 0 \tag{16}$$

$$0 \le (P_i^m - q_{i,h}^d) \perp \rho_{i,h} \ge 0 \tag{17}$$

$$0 \le (P_i^m - q_{i,h}^c) \perp \delta_{i,h} \ge 0 \tag{18}$$

$$0 \le x_{i,0} + \sum_{t=1}^{h} \left( \alpha q_{i,t}^c - \frac{q_{i,t}^d}{\beta} \right) - \frac{r_{i,h}}{\beta} \perp \gamma_{i,h} \ge 0 \tag{19}$$

$$0 \le C - x_{i,0} - \sum_{t=1}^{h} \left( \alpha q_{i,t}^c - \frac{q_{i,t}^d}{\beta} \right) \perp \varphi_{i,h} \ge 0 \tag{20}$$

$$0 \le -p_h^r + \lambda_{i,h} + \gamma_{i,h} + u_{i,h}' \perp r_{i,h} \ge 0 \tag{21}$$

$$0 \le -p_h^e + \rho_{i,h} - \frac{1}{\beta} \sum_{t=h}^{H} (\varphi_{i,t} - \gamma_{i,t}) \perp q_{i,h}^d \ge 0 \tag{22}$$

$$0 \le p_h^e + \delta_{i,h} + \alpha \sum_{t=h}^{H} (\varphi_{i,t} - \gamma_{i,t}) \perp q_{i,h}^c \ge 0 \tag{23}$$

where the expression $x \perp y$ means at most one of $x$ and $y$ can take a strictly nonzero value.

Conditions (16) to (20) are complementarity slackness conditions. Besides, conditions (21) to (23) are stationarity conditions. The inequalities on the left-hand side of (16) to (20) define, along with the nonnegativity definitions on the right-hand side of (21) to (23) the feasible space of the primal problem. The inequalities on the right-hand side of (16) to (20) define, along with the nonnegativity definitions on the left-hand side of (21) to (23) the feasible space of the dual problem. If $\left\{ r_{i,h}, q_{i,h}^c, q_{i,h}^d \right\}$ is feasible, it must be an optimal value of end user's problem.

It should be noted that constraints (16)–(23) are nonlinear and nonconvex, which makes the problem hard to solve. Under some reasonably mild assumptions, we linearize them by introducing additional binary variables. The following set of conditions can replace the former nonlinear ones:

$$0 \le P_i^m - r_{i,h} \le M z_{i,h}^1 \tag{24}$$

$$0 \le \lambda_{i,h} \le M(1 - z_{i,h}^1) \tag{25}$$

$$0 \le P_i^m - q_{i,h}^d \le M z_{i,h}^2 \tag{26}$$

$$0 \le \rho_{i,h} \le M(1 - z_{i,h}^2) \tag{27}$$

$$0 \le P_i^m - q_{i,h}^c \le M z_{i,h}^3 \tag{28}$$

$$0 \le \delta_{i,h} \le M(1 - z_{i,h}^3) \tag{29}$$

$$0 \le x_{i,0} + \sum_{t=1}^{h} \left( \alpha q_{i,t}^c - \frac{q_{i,t}^d}{\beta} \right) - \frac{r_{i,h}}{\beta} \le M z_{i,h}^4 \tag{30}$$

$$0 \leq \gamma_{i,h} \leq M(1 - z_{i,h}^4) \tag{31}$$

$$0 \leq C - x_{i,0} + \sum_{t=1}^{h} (\alpha q_{i,t}^c - \frac{q_{i,t}^d}{\beta}) \leq M z_{i,h}^5 \tag{32}$$

$$0 \leq \varphi_{i,h} \leq M(1 - z_{i,h}^5) \tag{33}$$

$$0 \leq -p_h^r + \lambda_{i,h} + \gamma_{i,h} + u_{i,h}' \leq M z_{i,h}^6 \tag{34}$$

$$0 \leq r_{i,h} \leq M(1 - z_{i,h}^6) \tag{35}$$

$$0 \leq -p_h^e + \rho_{i,h} - \frac{1}{\beta} \sum_{t=h}^{H} (\varphi_{i,t} - \gamma_{i,t}) \leq M z_{i,h}^7 \tag{36}$$

$$0 \leq q_{i,h}^d \leq M z_{i,h}^7 \tag{37}$$

$$0 \leq p_h^e + \delta_{i,h} + \alpha \sum_{t=h}^{H} (\varphi_{i,t} - \gamma_{i,t}) \leq M z_{i,h}^8 \tag{38}$$

$$0 \leq q_{i,h}^c \leq M z_{i,h}^8 \tag{39}$$

$$z_{i,h}^1, z_{i,h}^2, z_{i,h}^3, z_{i,h}^4, z_{i,h}^5, z_{i,h}^6, z_{i,h}^7, z_{i,h}^8 \in \{0, 1\}, \forall h \tag{40}$$

where $M$ is a positive constant that is large enough to guarantee that the inequalities are never binding when the right-hand is different from 0. As long as such an assumption holds and in view of the binary variable definitions in (40), the constraints (24), (25) are equivalent to (16) and so on for the constraints. We obtain the linearized KKT condition set:

$$U_i^{KKT} = \{(r_{i,h}, q_{i,h}^c, q_{i,h}^d, \rho_{i,h}, \lambda_{i,h}, \gamma_{i,h}, \delta_{i,h}, \varphi_{i,h}), \forall h \big| (24) - (40)\} \tag{41}$$

The corresponding dual function which minimizes the Lagrangian for the primal problem of end user is

$$\begin{aligned}
&\max D_i \\
&= \sum_{h=1}^{H} p_h^e l_{i,h} - (\lambda_{i,h} + \rho_{i,h} + \delta_{i,h}) P_i^m + (\varphi_{i,h} - \gamma_{i,h}) x_{i,0} - \varphi_{i,h} C \\
&s.t. \left\{ r_{i,h}, q_{i,h}^c, q_{i,h}^d, \rho_{i,h}, \lambda_{i,h}, \gamma_{i,h}, \delta_{i,h}, \varphi_{i,h} \right\} \in U_i^{KKT}
\end{aligned} \tag{42}$$

Based on strong duality theory, the optimal solution of convex problem equals to the optimal value of its dual problem (42). That is,

$$\begin{aligned}
&\sum_{h \in H} p_h^r r_{i,h} - u_{i,h} - p_h^e(l_{i,h} + q_{i,h}^c - q_{i,h}^d) \\
&= \sum_{h \in H} -p_h^e l_{i,h} + (\lambda_{i,h} + \rho_{i,h} + \delta_{i,h}) P_i^m - (\varphi_{i,h} - \gamma_{i,h}) x_{i,0} + \varphi_{i,h} C
\end{aligned} \tag{43}$$

Notice that aggregator's problem is complicated by the term $p_h^r r_{i,h}$. The objective function can be linearized by the duality theorem on the user's problem.

$$\sum_{h \in H} p_h^r r_{i,h} = \sum_{h \in H} (\lambda_{i,h} + \rho_{i,h} + \delta_{i,h}) P_i^m - (\varphi_{i,h} - \gamma_{i,h}) x_{i,0} + \varphi_{i,h} C + u_{i,h} + p_h^e(q_{i,h}^c - q_{i,h}^d) \tag{44}$$

Finally, the former problem (42) can equivalently be transformed into following optimization problem:

$$\begin{aligned}
&\max RPr - \sum_{i \in I} \sum_{h=1}^{H} (\lambda_{i,h} + \rho_{i,h} + \delta_{i,h}) P_i^m - (\varphi_{i,h} - \gamma_{i,h}) x_{i,0} + \varphi_{i,h} C + u_{i,h} + p_h^e(q_{i,h}^c - q_{i,h}^d) \\
&s.t. \left\{ r_{i,h}, q_{i,h}^c, q_{i,h}^d, \rho_{i,h}, \lambda_{i,h}, \gamma_{i,h}, \delta_{i,h}, \varphi_{i,h} \right\} \in U_i^{KKT} \\
&0 \leq R \leq \sum_{i \in I} r_{i,h}
\end{aligned} \tag{45}$$

where $U^{KKT}$ is the linearized KKT conditions of the user's problem. Now the proposed model is transformed into a MIQP, in which objective function is quadratic, and all constraints in problem are linear.

## 5. Simulation Results

In the above market settings, a large number of end users can be aggregated to participant in ancillary market via intermediary agent. We conduct simulations to examine the effectiveness of the proposed model. The model is formulated in MATLAB R2012a, and solved using CPLEX 12.7.

In this section, a distribution network containing 100 end users is considered. Assume that the aggregator can participant in reserve service bidding only if the bid is above 1 MW with the sustainable operating time of 4 h. The purchase prices are released to customers every 15 min. Because EV has its special characteristics of fast start up, it is assumed to response instantaneously.

Both the charging and discharging efficiency of EV is 90%, the maximum power rate is set to be 50 kW, the range of SoC is between 0% to 100%, the initial SoC is randomized. We define user-varying coefficients of disutility equation to model the differences of every end user. Limited by the electricity price policy in China, the price cannot fully reflect the supply and demand, which is inconsistent with the market scenario we discussed. The market-oriented operation of American electricity is relatively mature, e.g., NYUSO. Thus, price of NYISO has more reference value for analysis. Although the market policies vary from countries to regions, the methodology of this paper are meaningful and general in academic. The real time market price date which is published every 15 min comes from NYISO in US.

The optimal decisions of aggregator are shown in Figures 2 and 3. Consider the results in Figure 2, we can see the aggregator's bidding size and sustainable operating duration meet the market access requirement by collecting distributed resources. Response capacities in different colors are from different users. During daytime, for example 8:00 to 20:00, the aggregated reserve capacity is limited, because there are limited available EVs. While in night (00:00–8:00, 20:00–24:00), much more EVs corrected by distribution network are aroused for reserve service. Referring to Figure 3, the purchase prices in 8:00–20:00 are also high, encouraging end users to provide reserve capacity.

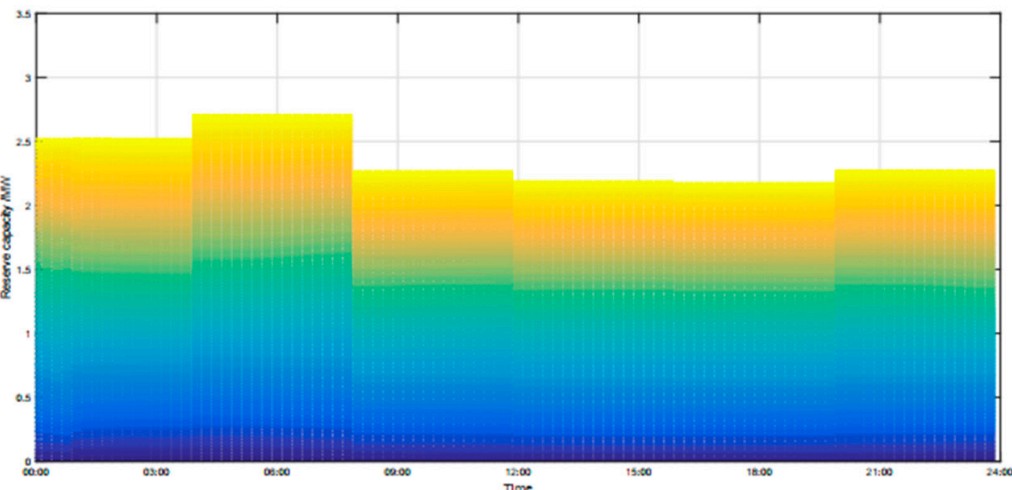

**Figure 2.** The aggregator's bidding strategy in wholesale ancillary market.

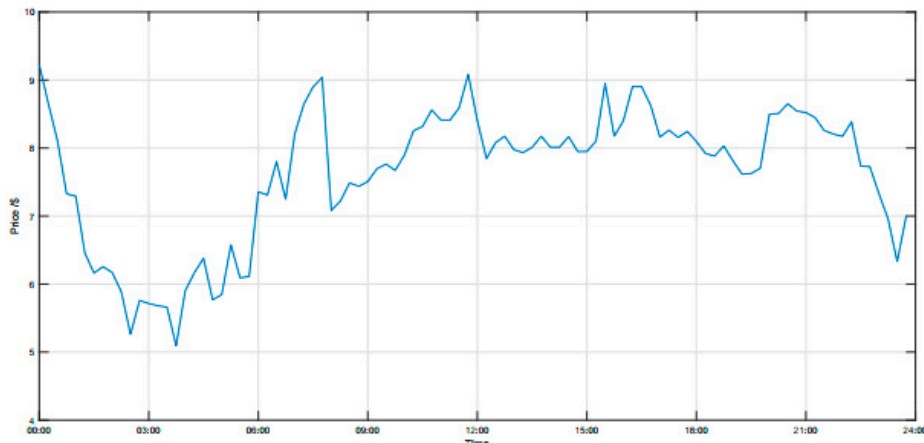

**Figure 3.** The aggregator's optimal purchase prices.

Note that the bidding sizes are defined as the minimum capacities in 4 h. In order to obtain an optimal portfolio of flat bidding size, the aggregator can adapt a strategy based on price control. The price control mechanism can be described as follows: firstly, the aggregator decides a price sequence, which contains 16 prices in 4 h. After receiving aggregated reserve capacity from users, the aggregator updates his behavior by raising the purchase prices in low-capacity period, while reducing the purchase prices in high-capacity period. The optimal price sequence is determined till the bidding size becomes flat. In this case, the aggregated reserve capacities are sold out in the wholesale ancillary market.

As shown in Figure 3, the resulting purchase prices of reserve change with high volatility during a day. The aggregator can obtain control of EVs by providing more attractive price. Consider an extreme case, if the purchase prices are high enough, the end user will spare no effort providing reserve services regardless of the satisfaction obtained by self-schedule. In reality, the purchase prices will not trend to be infinite, because the aggregator faces with a trade-off between investment and reward. It is understandable that the price goes up in the daytime while it goes down in the midnight. On the one hand, a proportion of EVs are not available in working hours during the day. On the other hand, the majority self-schedule demand of EVs lies in the peak hours.

Figure 4 shows the aggregator's cost, income and profit. It can be seen that the aggregator can earn the intermediate price difference in this transaction process. Besides, the aggregator gains higher profit when EVs are idle (00:00–8:00, 20:00–24:00). The result verifies the effectiveness of the model. The aggregator can effectively gather idle EV capacity to participate in the ancillary market as well as make profits.

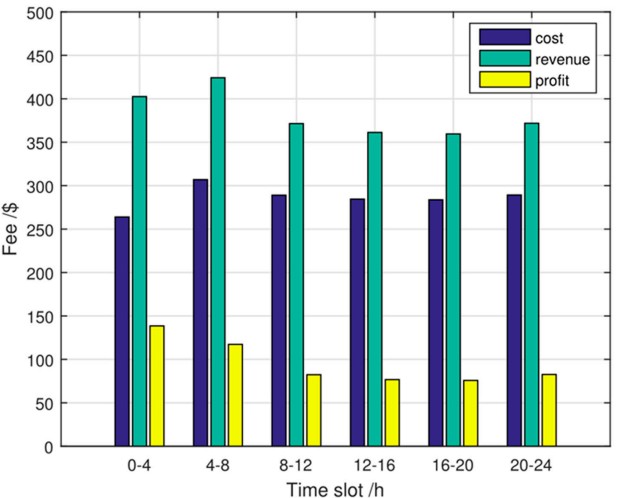

**Figure 4.** The aggregator's cost, income and profit.

## 6. Conclusions

In this paper, we study the optimal market participation of the aggregator, who coordinates private electric vehicles to provide reserve capacities in the wholesale capacity market. The hierarchical model is developed to model the interaction among aggregator, end user and the grid operator, in which the aggregator at the middle layer decides its price policy to purchase EV's capacity at the lower layer, and sells the aggregated reserve capacity to obtain payment from grid at the higher layer. Consider the strict access constraints of exiting capacity market, we also explore an incentive-based mechanism across different time scales (long time scales of the bidding interval and short time scales of EV's response interval). Then by jointly using KKT conditions, the disjunctive constraints and the primal-dual approach, the profit maximization problem can be reformulated into a MIQP. Finally, we adopt off-the-shelf software to solve this problem. The results show that the mechanism works well by benefiting both the aggregator and individual end user.

**Author Contributions:** Methodology, X.L. (Xianglu Liu); data curation, H.X.; writing—original draft, X.L. (Xianglu Liu) and H.X.; writing—review & editing, X.L. (Xianglong Li); supervision, H.C.; resources H.C.; project administration X.L. (Xianglu Liu), X.L. (Xianglong Li), W.Z. and Z.S.; funding acquisition, H.C. All authors have read and agreed to the published version of the manuscript.

**Funding:** This research was funded by Research on interaction between large-scale electric vehicle and power grid and charging safety protection technology (N.O. 5418-202071490A-0-0-00), Science and technology project of State Grid Corporation of China.

**Institutional Review Board Statement:** Not applicable.

**Informed Consent Statement:** Not applicable.

**Data Availability Statement:** The data presented in this study are available on request from the corresponding author. The data are not publicly available due to [information security]. Publicly available datasets were analyzed in this study. This data can be found here: [Historical pricing data of NYISO: https://www.nyiso.com/public/markets_operations/market_data/pricing_data/index.jsp].

**Conflicts of Interest:** The authors declare no conflict of interest.

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
