# Peer review of "Aggregate Electric Vehicles in Demand Side for Ancillary Service"

_wevj, doi:10.3390/wevj12040242_

Round 1

Reviewer 1 Report

In my opinion, the contribution of this research should be better explained, highlighting the contribution to the field under study. Accordingly, the methodology and the  application case should be clarified: how does the business model of EV aggregation work in this method? How loads would be physically aggregated and managed? What are the characteristics of the market where demand resources will be traded (the so-called here “ancillary wholesale market”)?

On the other side, Figure 1 is not understandable. What is the meaning of the different colors? What is the meaning of the different load levels during the represented day? Is there just one single profile? Why?

Finally, simulation results are not well explained: this section should be further developed and the obtained results should be discussed in more detail.

Reviewer 2 Report

This paper focuses on the participation of the electric vehicles to the wholesale capacity market via aggregators, especially on the aggregator’s behaviour using profit maximization problem. 
  • This paper collects data of end user by using the ‘Incentive Verification Experiment’ as explained in Section 2. However, it describes the coverage of the experiment and result without analysis process. Does Table1 show the measured actual result? If there has been additional data processing operation, please give a detailed information.
  • Is there reason to set up the simulation that the aggregator can participant in reserve service bidding only if the bid is above “1MW” with the operating time of “4hours”?
  • Why do the authors use price data from NYUSO which is in US not in China?

Round 2

Reviewer 1 Report

In my opinion, the ancillary services market proposed by the authors remains not being well defined and explained: It limits the understandability of the economic analysis performed in section 5, where the authors have evaluated a set of profits and revenues. After reading the new version of the manuscript, I do not see a clear explanation about how this market would work. What kind of products are considered (frequency regulation -primary, secondary, tertiary- voltage regulation, balancing services,…)? What size of power packages is considered? How trading would take place (pay-as-bid, marginalist auction, continuous trading,…)?. What prices are considered and how are they obtained? Has this market been designed for the purpose of this paper or the participation of EV aggregators in an existing market is considered? If so, what is this market? I think that all of these issues should be clarified.

Author Response

Thanks for your comment.

Firstly, a series of services necessary to maintain the stable operation of power system are collectively referred to as ancillary services, mainly including frequency stability, voltage stability, transient stability and so on. The key lies in power balance. The series products of ancillary services (frequency regulation, primary, secondary, tertiary  regulation, balancing services) are mainly divided by response time.  It has been proved that EVs can  provide a series of ancillary  services such as frequency modulation, demand response and ect. because of its ability of rapid response and accurate control. According to the response time and response effect, the prices of various ancillary services will be different, but they are applicable to the market model discussed in this paper. Therefore, this paper is not limited to a certain type of products.

Generally, it has certain requirements for the capacity scale and continuous operation time in the ancillary market.  We set the capacity scale as 1MW and the continuous operation time as 4H referring to the requirements of central Europe.

Furthermore, the market discussed in this paper is completely competitive, and the market price is determined by the marginal price. Although the horizontal competition relationship of multiple aggregators is also meaningful for further research. In this paper, we focus on the optimal decision-making of aggregators as well as users and  simplify the market competition model of multiple aggregators for the solvability of the mathematical model.

In addition, the market discussed in this paper is not fictitious. Some provinces and cities have allowed aggregators to participate in the ancillary  market. Unlike the existing market, which takes aggregators as price-take, aggregators act as an price-maker to bid on the premise of conforming to industry norms and the objective laws of power system. In the meantime, the interests of end users are well considered. All of these above are also the main contribution of this paper.